A systematic review on artificial intelligence approaches for smart health devices

http://orcid.org/0000-0003-2436-6835 Aversano Lerina 1
http://orcid.org/0000-0001-8025-733X Iammarino Martina 2 martina.iammarino@unipegaso.it
Mancino Ilaria 3
Montano Debora 4
1 Department of Agricultural Science, Food, Natural Resources and Engineering, University of Foggia , Foggia , Italy
2 Department of Computer Science, University of Bari Aldo Moro , Bari , Italy
3 Department of Engineering, University of Sannio , Benevento , Italy
4 CeRICT scrl—Regional Center Information Communication Technology , Benevento , Italy
Angiulli Giovanni
Electronic publication date: 2024 Oct 21
Publication date: 2024
Volume: 10
Electronic Location ID: e2232
Received 2024 Feb 8; Accepted 2024 Jul 12
Copyright: © 2024 Aversano et al.
Copyright year: 2024
Copyright holder: Aversano et al.
License: This is an open access article distributed under the terms of the Creative Commons Attribution License, which permits unrestricted use, distribution, reproduction and adaptation in any medium and for any purpose provided that it is properly attributed. For attribution, the original author(s), title, publication source (PeerJ Computer Science) and either DOI or URL of the article must be cited.
License URL: https://creativecommons.org/licenses/by/4.0/

Keywords: Artificial intelligence, Smart health, Wearable devices

Funding: The authors received no funding for this work.

==============================
In the context of smart health, the use of wearable Internet of Things (IoT) devices is becoming increasingly popular to monitor and manage patients’ health conditions in a more efficient and personalized way. However, choosing the most suitable artificial intelligence (AI) methodology to analyze the data collected by these devices is crucial to ensure the reliability and effectiveness of smart healthcare applications. Additionally, protecting the privacy and security of health data is an area of growing concern, given the sensitivity and personal nature of such information. In this context, machine learning (ML) and deep learning (DL) are emerging as successful technologies because they are suitable for application to advanced analysis and prediction of healthcare scenarios. Therefore, the objective of this work is to contribute to the current state of the literature by identifying challenges, best practices, and future opportunities in the field of smart health. We aim to provide a comprehensive overview of the AI methodologies used, the neural network architectures adopted, and the algorithms employed, as well as examine the privacy and security issues related to the management of health data collected by wearable IoT devices. Through this systematic review, we aim to offer practical guidelines for the design, development, and implementation of AI solutions in smart health, to improve the quality of care provided and promote patient well-being. To pursue our goal, several articles focusing on ML or DL network architectures were selected and reviewed. The final discussion highlights research gaps yet to be investigated, as well as the drawbacks and vulnerabilities of existing IoT applications in smart healthcare.

Introduction

The integration of artificial intelligence (AI) and wearable solutions has driven a significant transformation in the smart health industry. This dynamic synergy has captured the interest of both the scientific and industrial communities, promising revolutionary advances in healthcare and improved quality of life for people.

Specifically, wearable technology plays a central role in this transformation collecting and transmitting data in real-time, and promoting a more proactive approach to healthcare. Miniaturized sensors embedded in wearable devices can detect motion data to monitor human movements (Rastegari & Ali, 2020) and physiological and behavioral data, including heart rate (Rahman et al., 2020), blood pressure (Chiang, Wong & Dey, 2021), patterns of sleep (Shen et al., 2022), physical activities (Staab et al., 2022), SpO2 (Chou et al., 2020), among others. These data streams, along with patient records (Sabra, Mahmood Malik & Alobaidi, 2018), lifestyle and environmental data (Wu et al., 2022), provide an invaluable resource for both healthcare providers and individuals, offering insights into trends in health and enabling early detection of potential health problems. Additionally, some studies have leveraged Internet traffic from Internet of Things (IoT) devices (Yue et al., 2020) and social interaction data (Ware et al., 2022) to develop depression prediction techniques.

However, the process of collecting and transforming large amounts of raw data into useful information represents a significant challenge, mainly due to factors such as the risk of cyber-attacks and invasion of privacy (Bi, Liu & Kato, 2022), along with issues related to data overload and real-time sensory data processing latency (Gupta, Bhatia & Kumar, 2021). Furthermore, the substantial volume and complex nature of the data generated by wearable devices introduces a multitude of complexities that require the application of sophisticated analytical tools to transform it into meaningful information, and this is where artificial intelligence plays a critical role.

These technologies are proving indispensable for extracting valuable patterns, predicting health trends, and facilitating personalized health interventions. Their ability to adapt and learn from data allows the development of robust models that can continuously improve their accuracy and efficiency.

Therefore, our research strives to provide a systematic review of the current landscape where ML and DL are being leveraged to unlock the full potential of wearable applications in smart health. The analysis of the scientific literature aims to provide a comprehensive understanding of the prevailing trends, findings, and challenges in this rapidly evolving field. In this sense the following objectives have been defined, addressing five main goals: analyze the most recent and relevant research accomplishments in the field of AI applied to smart health;

investigate which of the two most well-known AI methodologies, machine learning (ML) and deep learning (DL), is primarily employed in the specific research topic;

evaluate the accuracy and reliability achieved by AI-driven models in the smart health landscape. This assessment provides crucial insights into the real-world feasibility and efficacy of these technologies in improving healthcare outcomes;

assess critical aspects related to security and data privacy;

outlines the research gaps still to be investigated.

In summary, through our systematic review, we aim to contribute to the current state of the literature, exploring the research landscape regarding ML and DL approaches applied in the smart health sector, to highlight challenges and vulnerability. These architectures form the backbone of the smart healthcare ecosystem, enabling the development of predictive models and decision support systems (Yin & Jha, 2017). In this way, we intend to highlight what future research efforts in this field should be.

In particular, we will examine the different types of DL architectures, evaluate the effectiveness and accuracy of these architectures in the context of healthcare applications, and review recent studies and research to understand their advantages and limitations. Additionally, we will address issues related to privacy and security of healthcare data in the age of AI. The increased use of AI technologies in healthcare raises concerns about protecting sensitive patient data and complying with privacy regulations.

The document is structured as follows. In “Related Work” we reported the related works, first analyzing the research articles that significantly influenced our study’s focus on the applications of artificial intelligence in wearable devices in the smart health sector and then the recent systematic reviews, placing the focus on the contribution of our work. In “Background”, we discussed the context related to the research topic and its current development. In “Research Method”, we outlined the search methodology employed by providing a comprehensive explanation of the search queries and selected databases, the search and filtering process, as well as presenting the distribution of analyzed articles by year and location. In “Results”, we report the results of our research work by classifying the selected articles based on the research questions outlined in the previous section. “Discussion” involves interpreting the results of our systematic review, discussing their implications, and analyzing any limitations or potential future research directions. Finally, “Conclusions” summarizes the conclusions of the research work.

Related work

In this section, we present the results of an analysis of research articles identified in the literature that address the topic studied. The analyzed research documents show different application areas, some clearly defined to focus on specific pathologies or healthcare applications.

The field of smart health encompasses a multitude of goals, each distinctly contributing to healthcare progress. These applications can be classified into primary domains which are diagnosis and classification (i), prediction (ii) and monitoring (iii). Studies focusing on diagnosis and classification delve into the accurate identification of various medical conditions. Among these, bipolar disorder (Hafiz et al., 2020) represents a significant challenge due to its cyclical nature and similarity of symptoms to other psychiatric conditions. Cardiac arrhythmia (Cheikhrouhou et al., 2021; Yin & Jha, 2017) requires particular attention due to its lethal potential and the need for timely interventions. Parkinson’s disease (Teng et al., 2023), a progressive neurodegenerative disease, benefits from early diagnosis to slow the progression of motor and non-motor symptoms. Type 2 diabetes (Yin & Jha, 2017) is a chronic metabolic condition that requires precise classification to effectively manage glycemic control and prevent complications. Urinary bladder disorders (Yin & Jha, 2017), often characterized by common urological symptoms, require an accurate differential diagnosis to identify potentially malignant conditions. Lung cancer (Masood et al., 2018) diseases are among the most lethal, and early diagnosis is crucial to improve the chances of survival. Finally, breast cancer (Yu et al., 2021), one of the most common forms of cancer among women, requires advanced classification techniques to personalize treatments and improve clinical outcomes. Some studies focus on advanced and innovative methods of integrating and analyzing data from various sources to enhance the accuracy of medical diagnoses through machine learning, thereby improving the predictive capability of AI models in the early diagnosis of diseases (Singh et al., 2021). These applications aim to improve the accuracy and efficiency of medical diagnoses, helping healthcare professionals provide better care to patients. Another critical aspect is the prediction of health-related risks and conditions. This includes predicting the risk of diseases such as COVID-19 (Neog, Dutta & Medhi, 2022), heart disease (Chakraborty & Kishor, 2022; Sarmah, 2020), viral infections (Skibinska et al., 2021) Parkinson’s disease (Goñi et al., 2022), diabetes (Xiao et al., 2022; Ferdousi, Hossain & Saddik, 2021) and even exposure to an epidemic (Ng et al., 2022). Furthermore, the application of deep learning techniques for analyzing complex biometric signals is examined, significantly enhancing the ability to detect anomalies that precede critical clinical events, and it is demonstrated how advanced data processing can improve prevention in healthcare settings (Zhong et al., 2023). Other studies develop prediction systems for health conditions such as blood pressure (Chiang & Dey, 2019) and flat feet (Kim et al., 2020). These predictive models enable both individuals and healthcare systems to take proactive measures for prevention and early intervention. AI-powered monitoring systems offer real-time insights that can help people and healthcare providers make informed decisions. The ability to continuously monitor vital signs and physiological parameters such as respiratory rate (Shuzan et al., 2021), tidal volume (Schoun et al., 2018), and brain oxygenation (Chou et al., 2020) is vital for the management of chronic diseases and ensuring general well-being. Furthermore, a distinct set of research articles highlights the indispensable role of wearable devices that facilitate comprehensive health monitoring, cognitive assessment, and monitoring of conditions such as myotonic dystrophy type 1 (Chapron et al., 2021), heart disease (Ramkumar et al., 2023) and nocturnal diuresis (Kuru et al., 2024). Frequently, research involves the development and design of the wearable device, as exemplified by the MyPad device (Kuru et al., 2024). However, smartwatches and fitness trackers are also employed to detect conditions such as respiratory disorders (Chen et al., 2021), asthma (Wu et al., 2023), and depression (Chikersal et al., 2021). Numerous studies have emerged demonstrating the development of multi-objective systems that integrate monitoring, classification, and prediction capabilities, as they not only monitor various parameters and conditions in real time but also classify and predict outcomes based on the monitored data (Bellos et al., 2014; Prathaban, Balasubramanian & Kalpana, 2021; Sopic et al., 2018). The importance of scalability and adaptability of AI models in handling large volumes of real-time generated health data is emphasized, highlighting the need for computationally efficient models for long-term monitoring applications (Singh et al., 2023). Additionally, some studies underscore the significance of integrating predictive analytics and real-time monitoring to optimize the management of home healthcare (Adhikary et al., 2023). Therefore, the combined efforts in these different fields illustrate the multi-faceted potential of artificial intelligence and wearable technology to revolutionize healthcare diagnosis, prediction, screening, monitoring, and patient care.

Regarding the contributions of other recent systematic reviews related to artificial intelligence applied to smart wearable devices, or the healthcare sector, we have examined some of them to highlight their differences compared to the proposed investigation. For example, Site, Nurmi & Lohan (2021) focus primarily on the application of machine learning algorithms in the analysis of data from wearable devices in eHealth, and the most notable limitation of this study is a significant lack of exploration of the use of DL techniques for smart wearable devices in the smart healthcare sector. The studies Ahmadzadeh, Luo & Wiffen (2022) and Alam et al. (2018) do not adhere to the Kitchenham guidelines for systematic reviews (Wu et al., 2022), as they do not offer details on the research databases used, the search queries performed, the exclusion and inclusion in the filtering process or the number of documents discovered. Mughal et al. (2022) delve into the use of wearable sensors for the management of pathological conditions underlining their usefulness, challenges, and results. Alam et al. (2018), however, specifically focus on the integral role of communication technologies in supporting IoT applications related to the healthcare sector. Both research articles do not investigate artificial intelligence methodologies, including ML and DL, as they concern the analysis of data obtained from wearable devices in the field of smart health. Instead, the study (Qaim et al., 2020) systematically examines energy efficiency considerations in the field of wearable devices integrated with the Internet of Things and incorporates the shortcomings identified in the previously mentioned studies.

This research study proposes a systematic review, according to the guidelines of the Kitchenham method (Wu et al., 2022), to explore the research landscape regarding the application of artificial intelligence, with its ML and DL techniques, in the smart health sector.

Background

Artificial intelligence techniques

Artificial intelligence is a field of computer science that aims to develop systems capable of performing tasks that normally require human intelligence (Winston, 1984). These tasks include reasoning, learning, sensory perception, understanding natural language, and interacting with the surrounding environment. The goal of AI is to create machines and algorithms that can emulate or surpass human capabilities in specific domains.

It is divided into two large classes (Wang & Siau, 2019). The first represents weak artificial intelligence, also known as specialized AI, which refers to systems designed to perform specific tasks without necessarily understanding or imitating human intelligence in a general way. Examples include search engines, recommendation systems, and gaming algorithms. The second is strong artificial intelligence which aims to reach a level of intelligence comparable to human intelligence. In contrast to weak AI, strong AI should be able to tackle any intellectual task. However, currently, strong AI is still largely a future goal and subject to debate.

There are four main generic approaches, ML and DL, natural language processing, which focuses on the ability of computers to understand, interpret, and generate human language, and computer vision which is the ability of machines to interpret and analyze visual data, often used in applications such as facial recognition, robotic vision and medical image analysis (Dargan et al., 2020; Adam & Mukhtar, 2024). AI is an ever-evolving discipline with a growing impact on multiple industries, including healthcare, industrial automation, transportation, marketing, and many others.

More specifically, the application of AI in medicine is revolutionizing the healthcare sector, offering opportunities to improve the diagnosis, treatment, and management of patients. The main application areas are medical diagnosis where machine learning algorithms, particularly convolutional neural networks (CNN), are used to analyze medical images such as X-rays, MRIs and computed tomography scans to identify abnormalities and aid in the early diagnosis of diseases like cancer; clinical decision assistance, where machine learning algorithms are used to provide personalized recommendations to doctors, based on clinical data and test results, helping in choosing the most appropriate treatment options; treatment personalization where AI helps identify subgroups of patients with similar genetic or molecular characteristics or analyze historical patient data to predict treatment responses and optimize treatment protocols; and finally healthcare management and continuous monitoring, where AI facilitates the provision of remote medical care, allowing continuous monitoring of patients, management of chronic diseases and virtual medical consultancy, and introduces wearable devices that integrate with algorithms of AI can collect data on vital signs and report any anomalies, improving proactive health management (Kavitha et al., 2023; Al Kuwaiti et al., 2023).

Machine learning

ML is a branch of AI dealing with the development of algorithms and models that allow computers to learn from data and improve their performance without having been explicitly programmed (Mitchell, 1997). Instead of following explicit instructions, machine learning models draw insights from data so they can make decisions or make predictions based on new information.

The fundamental goal of machine learning is to allow computers to autonomously learn and adapt to new data and situations, continuously improving their performance over time. This is achieved by identifying patterns and relationships in the training data, which are then used to make predictions or decisions on previously unseen data.

There are three main types of learning in the context of machine learning: supervised learning, unsupervised learning, and reinforcement learning. In the first, the model is trained on a labelled dataset, where each input example is associated with a corresponding label or desired output. The goal is for the model to learn to map input features to desired outputs. Among the most used algorithms we find support vector machines (SVM) which try to find the best hyperplane that separates the data into different classes in the feature space (Noble, 2006); the K-nearest neighbors (K-NN) which classifies a data point based on the majority of labels of its nearest neighbours in the training set (Peterson, 2009); decision trees and Random Forests which recursively divide the data set into subsets based on decision criteria; and boosting algorithms that combine several weak models to create an overall stronger model. The idea is to iteratively train weak models, giving more weight to mistakes made earlier, so that the final model gives more attention to the more difficult cases. In unsupervised the model is trained on unlabeled data and tries to identify intrinsic patterns or structures in the data without having a specific output to predict (Barlow, 1989). The main goal is to identify hidden relationships or groupings in the data. Finally, with reinforcement learning the agent learns to make decisions by interacting with an environment. It receives feedback in the form of rewards or penalties based on actions taken, guiding the improvement of its strategies over time. Some examples of machine learning applications include facial recognition, machine translation, medical diagnosis, text classification, spam filtering, and many others. The advancement of computer technologies, accessibility to data and increasingly powerful computing capabilities have contributed to the growing importance and diffusion of machine learning in various sectors (Jiwani, Gupta & Whig, 2023).

Deep learning

Machine learning, which is a neural network with three or more layers, includes deep learning as a subset. These neural networks allow the system to “learn” from large volumes of data by simulating the functioning of the human brain, although far from reaching its potential (Deng & Yu, 2014). Although a neural network with a single layer is small, it is capable of approximate predictions, while adding additional hidden layers can support the optimization and improve accuracy. Instead, for deep learning deep neural networks (DNN) try to mimic the human brain by a combination of data inputs, weights, and distortion. These components cooperate to precisely identify, classify, and characterize features and information inside the data. A DNN is made up of several layers made up of interconnected nodes; each layer builds on the results of the one before it to refine and improve the final classification and prediction. Forward propagation is the term used to describe this elaboration process as it progresses through the DNN layers. A DNN’s visible layers are the first and last layers, which designate the input and output layers, respectively. The neural network gathers data for the subsequent stages of the analysis in the first layer, known as the input layer, and elaborates the final prediction or classification in the last layer, known as the output layer.

Specialized architectures are CNNs (O’Shea & Nash, 2015) and recurrent neural networks (RNNs) (Medsker & Jain, 2001). CNNs can identify characteristics and patterns within an image, enabling tasks like detection and recognition. CNNs are largely utilized in computer vision and image classification applications. In 2015, a convolutional neural network beat a human in an object recognition challenge. RNNs are typically used in natural language recognition and speech recognition applications as they leverage sequential or time series data.

Smart health

E-health refers, generically, to the various processes of digitalization of healthcare and, therefore to the creation of electronic infrastructures for the administrative and not just clinical management of each citizen’s data, smart health focuses more on IoT devices, including smartphones, which implement those same processes and personalized models for monitoring and treating health conditions (Tian et al., 2019).

Smart health is the basis of the so-called 4P medicine: Preventive, Participatory, Personalized, and Predictive. It is a medicine capable of preventing the disease based on the intersection of the different biological profiles of the person, through a simple blood sample and/or the processing of data from IoT devices (Aversano et al., 2022a; Chang, 2023). A medicine capable of making each person no longer a “patient” but aware and involved in their state of health thanks to real-time communication with their doctors. But also to personalize diagnosis and treatment (Aversano et al., 2021; Adam & Mukhtar, 2024) based on the risk factors detected and the information received from the wearables and to predict the severity of the risk factors of pathology before it manifests itself.

Smart health serves to automate procedures and simplify bureaucracy, contain costs, reduce times, optimize flows, make people aware of their health condition through the transparency of data, immediately consultable, and develop “person-friendly” health paths. It is essential for monitoring the quality of air, water, and building safety.

Smart health allows you to be assisted almost everywhere, realizing the famous transition from expectant medicine to initiative medicine, or from an emergency model to a proactive model, of reticular, “care-intensive” assistance, which sees the hospital as a high-intensity centre for acute pathologies, and the “low-intensity” territorial facilities as structures that manage chronic conditions, prevention and social-welfare services (Aversano et al., 2022b; Mshali et al., 2018).

Wearable devices

The term wearable devices refers to a wide range of intelligent devices that can be connected to other electronic devices such as smartphones, wirelessly or via Bluetooth technology, allowing not only the detection but also the storage and exchange of data of different like, immediately and often without the need for human intervention. Increasingly advanced, wearable devices are used for various purposes: from the detection of biometric signals, through sensors positioned on the skin, to quick access to online information (Iqbal et al., 2021; Devi et al., 2023; Tariq, 2024). Among these, we find wearable devices such as heart rate monitors, glucometers, or sphygmomanometers which are now able to offer a valid contribution when used in the healthcare sector. In fact, through sensors, actuators, and software connected from the device to a smartphone or tablet with the cloud, intelligent wearable devices worn by patients allow the collection, analysis, and real-time transmission of personal health data (Babu et al., 2024; Cui, Du & Wu, 2023).

Wearable healthcare is based on the collection of real-time or continuous 24-h data on a person’s state of well-being and physical fitness (Cho, 2019; Escobar-Linero et al., 2023). This data allows you to create a personalized health database. Anyone who wears a smart device capable of detecting health data can therefore use it to allow a healthcare worker to collect their data to obtain remote monitoring and to achieve health, fitness, well-being, and physical fitness objectives that are not necessarily related to the control of pathological conditions.

In healthcare, wearable devices such as healthcare smartwatches are often used to monitor patients’ health conditions and related emergencies, allowing them to be recognized as soon as they occur. The use of wearable technology therefore creates the potential for a proactive approach to healthcare. This can help detect symptoms of an illness before they develop into larger problems and, therefore, can have dangerous health consequences. Even people with known pathologies can benefit from early identification of signs of possible anomalies.

A typical example of a smart (i.e., intelligent) device for health monitoring is smart patches (Segev-Bar, Konvalina & Haick, 2015; Wong et al., 2023). These are thin stickers containing electronic components (including sensors and actuators with appropriate processing, energy storage, and communication capabilities) that collect a patient’s vital signs: heart rate, body temperature, and more, and transfer them to appropriate apps for doctors and patients.

Another example is a mobile ECG monitor (Baig, Gholamhosseini & Connolly, 2013). This device records the electrocardiogram via a wireless electrode that can be applied, for example, to the patient’s chest or finger. The ECG data is sent to the cloud and, then, to a doctor for appropriate evaluation.

Therefore, wearable technologies can represent innovative and highly effective solutions for patient health, while also offering enormous potential benefits to healthcare workers and the entire healthcare system (Tariq, 2024). But, to improve the usability and functions of the devices for practical use, issues such as acceptance by all users, security, ethics, and issues relating to the management of the big data generated still need to be addressed. from wearable devices. There are also elements of concern that arise from a large-scale wearable implementation, such as aspects of data privacy and security, but also supply chain management (Ahmed et al., 2024). When it comes to wearables and security, the medical Internet of Things opens the door to new attack vectors, including targeted malware and distributed denial of service threats (Shanmugam & Azam, 2023). Therefore, to avoid privacy and security issues, it is necessary to establish robust authentication and authorization processes for collecting, sending, and analyzing data.

Research method

In this section, we detail the methodology followed to conduct the systematic review, in terms of research questions, databases analyzed, and filtering and search methods used. Specifically, we based our research approach on the principles outlined by Kitchenham (2004) who proposed a guideline for systematic reviews. In detail, it consists of the following sequential phases: definition of relevant research questions;

definition of appropriate search queries based on extracting keywords from search questions;

selection of databases in which to search;

definition of initial filtering criteria such as search time interval, quality of searched results, etc.;

screening of titles and abstracts to exclude irrelevant and duplicate articles;

definition of more in-depth eligibility criteria to apply them during the in-depth reading of the definitively selected documents;

analysis of the remaining articles based on the research questions defined at the beginning.

Goal of the study

The focus of our work on systematically reviewing applications of AI in smart health is motivated by the growing importance of this research area and its significant implications for improving healthcare services and patient well-being. With the increasing adoption of digital technologies in the healthcare sector, the use of AI is becoming increasingly relevant to address complex challenges related to the diagnosis, treatment, and management of diseases. Therefore, our work aims to provide a detailed and up-to-date overview of AI applications in this emerging context.

The evolution of AI technologies, such as ML and DL, has opened up new opportunities to develop innovative and personalized solutions in the healthcare sector. By exploring the applications of these technologies, we aim to identify best practices and emerging trends in smart health. Understanding how AI can be used to optimize early diagnosis, improve chronic disease management, optimize care processes, and provide decision support to healthcare professionals is crucial to promoting the quality and effectiveness of healthcare services.

Our main objective is to conduct a systematic review to thoroughly explore the applications of AI in smart health. Through this review, we aim to not only evaluate the effectiveness and accuracy of existing AI solutions in improving clinical outcomes and optimizing operational efficiency in healthcare services but also to examine the current challenges and limitations in implementing AI in smart health-related privacy issues that need to be addressed.

Research questions and queries

A systematic review summarizes existing work allowing the completeness of the research to be assessed. The main advantage is that it provides information on the effects of the phenomena under observation. About that the purpose of this study is to explore the state of the art in integrating AI with IoT wearable devices in the field of smart health. In this regard, four research questions were defined.

First, the choice of AI methodology to apply to data collected by wearable IoT devices is crucial to the success of smart health applications. In this context, it is crucial to understand the complexity and depth of the AI models used, as well as the amount of data required for their correct functioning. The complexity of AI models, such as DL, involves the use of deep neural networks with many layers and parameters. This complexity allows the model to capture and learn sophisticated patterns in the data, improving predictive and analytical capabilities, but it is widely known that DL models require large amounts of data to train effectively. In contrast, traditional ML methods can work well even with smaller datasets. These methods are often less complex and require less data to provide accurate and reliable results. Therefore, in this study we aim to evaluate which is the most suitable approach for wearable IoT applications in the field of smart health, addressing our first research question:

RQ1: To what extent research studies adopt ML or DL approaches on smart wearable data?

The field of DL offers a wide range of architectures, each with specific characteristics and advantages. Data collected by wearable IoT devices presents unique challenges, such as temporal variability, the need for real-time analytics, and managing large volumes of data. It is essential to identify which DL architectures are best suited to make the most of this data and provide accurate and useful results. Examining which DL architectures are most commonly used in studies and applications related to wearable IoT devices helps identify trends and best practices in the field. In this regard, we address our second research question:

RQ2: Considering the scopes of application, which Neural Network architectures are more effective to support IoT Smart Health ?

In healthcare applications, the accuracy of algorithms is critical. Clinical decisions based on incorrect analyses can have serious consequences for the health of patients. Therefore, it is crucial that the algorithms used can provide accurate results. At the same time, the reliability of the algorithms ensures that the results are consistent and reproducible. In a healthcare context, where decisions must be based on robust and reliable data, trust in the results produced by algorithms is essential. Therefore, we address the third research question:

RQ3: Which algorithms lead to better performances while exploiting Smart Health data collected from IoT device monitoring?

Finally, health and medical data is extremely sensitive and personal. This data includes detailed information about individuals’ physical and mental health, which, if exposed, could cause significant harm, such as discrimination, invasion of privacy, and reputational damage. The management of healthcare data must comply with strict regulations, which impose high standards of data protection. Addressing our fourth research question allows us to identify and implement best practices and technologies to protect sensitive data, thereby improving the security and effectiveness of smart healthcare applications.

RQ4: RQ4: What privacy and security issues have taken into account while using health data collected from IoT devices?

Once the main study directions had been identified, the above-mentioned research questions were transformed into a specific query. This query was used to examine the selected databases, as described in the next paragraph, to find studies to be considered in the review. Below is the query used:

Q: (Wearable) AND (Artificial Intelligence) AND (Smart Health)

Databases

In this paragraph we report the databases that we queried with the query just described, to extract the articles on which our analysis focused. Below are the databases: IEEEXplore (https://ieeexplore.ieee.org/Xplore/home.jsp): a digital library and research database for discovering and accessing journal articles, conference proceedings, technical standards, and related materials in computer science, electrical engineering, electronics, and related fields.

ScienceDirect (https://www.sciencedirect.com/): which provides access to a large bibliographic database of scientific and medical publications from the Dutch publisher Elsevier.

The ACM Digital Library (https://dl.acm.org): a repository of published resources in the computing field maintained by the Association for Computing Machinery.

Research and filtering process

Having defined the research directions and the databases from which we have extracted the studies to be considered for the analysis, we can now tackle the search and filtering process that we have conducted. Specifically, this is shown in Fig. 1. It can be noted that the process consists of three substantial phases. The first consisted of applying the query to the chosen databases, while the second involved initial filtering of the selected documents based on the defined inclusion (IC) and exclusion criteria (EC), reported in Table 1. Finally, the third phase consisted of the actual analysis of the studies considered for the review.

Figure 1 Research and filtering process for the article selection.

Table 1 Inclusion and exclusion criteria applied in the research process.

Acronym	Description of the criterium	
Inclusion criteria		
IC1	Studies written in English	
IC2	Studies published in the range 2014–2023	
IC3	Studies published in a press or in a journal	
IC4	Studies should use ML or DL for Smart Health in IoT	
Exclusion criteria		
EC1	Studies are literature surveys or systematic reviews	
EC2	Studies do not entail IoT	
EC3	Studies are not related to Smart Health	
EC4	Studies are not specifically focused on ML or DL	

As shown in Table 1, the search process involved the selection of articles published in the year range from 2014 to 2023 (IC2) from the databases listed in “Databases”, using the keywords of inputs consisting of queries (Q) defined in “Research Questions and Queries”. This time horizon was chosen because, before 2014, the literature lacked relevant publications regarding ML or DL approaches in smart healthcare. This first phase of the search produced a total of 292 articles, of which 107 came from IEEEXplore, 121 from ScienceDirect and 64 from ACM.

The research work continued with the filtering process through (i) reading the titles, (ii) scanning the abstracts, and (iii) removing duplicate documents, thus arriving at a total of 105 selected articles. The selected studies had to fully meet the eligibility criteria in the Table 1.

In particular, they had to be written in English (IC1), published in the period 2014–2023 (IC2), they had to be focused on an IoT scenario (EC2), was to be specifically focused on DL or ML approaches (EC4) in the smart health domain (IC3) and was to be neither a survey nor a review (EC1). Furthermore, the search was limited to articles published or in press (IC1), thus excluding conferences and preprints. This decision was guided by the quality standards we aimed for in our systematic review. We considered that preprints have not been peer-reviewed and that solid and valid conference contributions are usually published in journals in more detail. Therefore, this final filtering step allowed us to retain a total of 58 articles to analyze across the defined research questions.

For completeness, Fig. 2 presents the temporal distribution of the works definitively considered. This shows the years on the x-axis and the number of contributions on the y-axis, showing the total number of contributions with the blue bars, and the percentage with the orange bars. As can be seen from the figure, the number of articles addressing our research topic has become significant since 2017 and has increased over the years, except for 2019.

Figure 2 Yearly distribution of the analyzed articles.

Furthermore, in Fig. 3, we show the distribution of the articles considered per publisher, always reporting the name of the publisher in abscissa, the number in the ordinate axis. In particular, the blue bars consider the number of articles published by the editor, instead of the orange ones percentage. The analysis of the figure indicates the predominance of the articles of the IEEE publisher related to the topic of our research.

Figure 3 Distribution of the articles per editor.

Results

In this section, we report the results obtained for each research question. Most of these results will be illustrated through grouped bar graphs to represent two series of distinct values in the same graph: the blue bars represent the absolute frequency of the number of cards. In contrast, the orange bar represents the relative percentage value. Where this type of graphic representation cannot be used given the nature of the specific analysis, the relevant graph will be explained together with the inherent results.

RQ1: To what extent research studies adopt ML or DL approaches on smart wearable data?

Different methodologies and approaches can be used to analyze health data. Among these, ML and DL algorithms and networks are the best-known and most used in the literature. There are the most disparate ML and DL techniques and there is also no shortage of customized implementations of basic algorithms, which makes the range of possible solutions truly wide.

For this reason, first of all, it is important to investigate which methodology is most used in the selected studies to determine the complexity and depth of the model with a distinction between ML and DL approaches or to investigate whether the study investigates both methodologies, therefore considering, in this case, the number of ML/DL algorithms evaluated.

Figure 4 summarizes the absolute number of articles (in blue) and mass percentages (in orange) for each analysis method used in the study (ML, DL, or both algorithm types) plotted on the x-axis. It can be observed that the majority of articles (57%) concern ML methodology, followed by DL algorithms (33%). However, only in 8 studies (10%), the two approaches are used together.

Figure 4 Distribution of the use of ML and DL techniques in the selected studies.

In Fig. 5 we report the boxplot of the distributions of the number of algorithms used in the selected articles, specifically the distribution of the ML classifiers evaluated in the selected articles is highlighted in blue boxplot and that refers to the DL algorithms is highlighted in orange one. In particular, Fig. 5A shows the distributions of the number of ML/DL classifiers evaluated in the studies that only used the ML or DL approach; Fig. 5B presents the distributions of the ML/DL classifiers evaluated in the studies that used both methodologies.

Figure 5 (A) Distributions of the number of ML/DL classifiers evaluated in the studies. (B) Distributions of the number of ML/DL classifiers evaluated in the studies that used both methodologies.

The analysis of the distributions of the number of ML/DL classifiers evaluated in the studies that used only of approach shows that in the majority of articles using ML algorithms, it is preferred to compare multiple classifiers of this type, at least more than 3 (median of the blue distribution). Instead, in studies where DL methodologies are used, it is preferred to use a single network selected according to the nature of the data. This choice is assumed to be made because DL algorithms require longer execution times and more computational memory than ML classifiers. Even in the case of articles that used the two approaches together, more ML classifiers are used than the DL approach which remains at the level of single network used.

According to this research question, we also explored some aspects related to the data set used in the selected articles. Firstly we examine the possibility of accessing the data, specifically if the data set used is public or private/personal. Figure 6 displays the data access policy adopted by the authors. The majority of the articles analyzed ( 67%) operate with private/personal data sets, while 33% of them analyze public data that in the majority of the cases are publicly available or can be released on request.

Figure 6 Distribution of the considered articles according to the data access policy.

Relating to the nature of the data used in the considered articles, we also investigated the number of instances/patients involved in collecting the data. Additionally, we also scrutinized the number of features used to train the models used in the collected articles. In Table 2 we disclose the results of this investigation. In particular, in it, we have reported in line with the confidence interval of the average number, the modal value, the minimum and maximum value assumed by the number of instances/patients involved in the data collection (first column), and the features used in the model training (second column).

Table 2 Metrics relating to the analysis of the data used in the considered articles.

Metrics	N. of instances/patients	N. of features	
Mean	1,631	63	
Standard deviation	5,925	148	
Minimum	2	2	
Maximum	42,021	773	

Referring to the results obtained in Table 2, we notice that for both analyses the minimum and maximum values differ greatly and for this reason, the mean and its relative confidence interval are extremely high; this is also the reason why we decided to study the modal value of the two distributions. In particular, taking this last value into account, we note that in general, the number of instances/patients in the articles is around 40 cases while the number of features used to train the models in the majority of the articles is 16.

RQ2: Considering the scopes of application, which neural network architectures are more effective to support IoT smart health?

Neural network architectures play a significant role in IoT Smart Health by enabling the development of applications that improve health monitoring, diagnostics, and overall well-being of users. Furthermore, DL architectures, together with appropriate data pre-processing and model optimization techniques, contribute to the advancement of IoT smart health applications, enabling early disease detection and better management of chronic conditions. In the documents analyzed, we found that there are some key neural network architectures used in IoT smart health concerning specific types of data: CNNs are widely used for image-based diagnostics for example analysis of medical imaging data (X-rays, MRI, CT scans), and real-time video monitoring (e.g., for patient movement analysis). They help in identifying patterns, anomalies, and potential health issues from visual data.

RNNs are often used for time series analysis of health-related data, such as continuous monitoring of vital signs (heart rate, blood pressure)and patient activity tracking.

Long short-term memory networks (LSTMs), an extension of RNNs, are valuable for processing and analyzing sequential health data, such as medical history, and time-stamped sensor data.

Densely connected convolutional networks (DenseNet) are a CNN architecture that connects each layer to every other layer in a feed-forward pattern. They are efficient for medical image analysis, especially when dealing with limited data or for transfer learning tasks.

Dense neural networks (DNNs) are often used in IoT Smart health for remote patient monitoring. DNNs can analyze data from wearable devices, sensors, and other IoT-enabled medical devices to continuously monitor a patient’s vital signs, detect anomalies, and provide real-time alerts to healthcare professionals for early intervention.

Concerning the pathology analyzed, we report in Fig. 7 the distribution of the articles we considered according to the disease having been tackled.

Figure 7 Distribution of pathologies analyzed in the selection of articles.

In particular in Fig. 7 the blue bars take into account the number of studies, while the orange bars report the the percentage; the same figure also shows the table with the respective values. The results of this analysis show that the most studied diseases are cardiac and neurodegenerative diseases (24%), followed by lung diseases (20%), following mental diseases such as depression and panic attacks (12%).

In Fig. 8 we have reported the distribution of the pathologies analyzed only with a DL approach. Also in this figure, we have displayed the number of studies in blue and the relative percentage in orange; at the bottom of the image we report the table of the relative values of both. This analysis shows that complex approaches such as DL are most used when dealing with heart and neurodegenerative disease data (20%), but also for mental health and lung disease data (15%). In particular, these diseases are the most difficult to diagnose because they present vast symptoms and therefore many characteristics that need to be managed during the analysis and classification of patients affected by them and all these aspects justify the use of complex techniques such as neural networks.

Figure 8 Distribution of pathologies analyzed with a DL approach.

To understand which architectures are the most implemented in the case of medical data collected by IoT intelligent devices, we analyze the distribution of the most used DL approaches in article selection. In Fig. 9 we present the distribution of the type of neural network architecture used in the selected articles. In particular, the card numbers are shown in blue, while the percentages are considered in orange; furthermore, at the bottom of the figure, there is a table of the values assumed by the coloured bars. The results show that the most used neural network architectures for IoT smart health problems are the convolutional neural network, the deep neural network, and the short-term memory network: in fact, these three different architectures alone cover more than 60% of the architectures used.

Figure 9 Number of articles and relative percentage, based on the type of neural network architecture.

For completeness, to understand which neural network architectures are used in IoT smart health considering the pathology, we carried out a cross-analysis between the type of neural network and the pathology, the results of which are presented in Fig. 10. The analysis shows that CNNs are the most used when dealing with clinical data relating to neurodegenerative diseases, mental problems, diabetes, heart problems, and cancer. Furthermore, DNNs are more used for walking problems, lung diseases, and neurodegenerative diseases, while LSTMs are used when the data is related to heart problems, COVID-19, mental health, lung diseases, and walking problems.

Figure 10 Distribution of neural network architectures based on the pathology analyzed by the different types of neural network architectures.

RQ3: Which algorithms lead to better performances while exploiting smart health data collected from IoT device monitoring?

ML and DL algorithms play a vital role in improving the accuracy of the results produced and the reliability of healthcare data collected through IoT device monitoring. Algorithms can be trained using historical data to identify patterns and trends in healthcare data. Therefore, their application can be particularly useful for continuous monitoring of vital parameters such as blood pressure, heart rate, or blood sugar levels, or for long-term data analysis, allowing the identification of correlations between different parameters and risk factors. This can help develop personalized treatments and targeted preventive strategies, thus optimizing the effectiveness of treatments and improving patients’ quality of life. For all these reasons, the ML or DL methodologies used to analyze the data must provide truly accurate results.

Consequently, we study the distributions of accuracy metrics obtained with the different approaches proposed in the selection of the analyzed documents to find out which algorithms bring better and more accurate results in the classification of healthcare data collected by IoT devices.

In this regard, we conducted a quantitative analysis, the results of which are reported in Fig. 11, which shows the distribution of the values of the accuracy metrics reported in the selected articles using the different approaches used. In the 58 studies analyzed, the minimum accuracy value obtained is 68% using multinomial logistic regression, while the maximum accuracy obtained is 100% with Cubic SVM. On average, studies achieve an accuracy of 88% (confidence interval of the mean [79.80–96%]).

Figure 11 Distribution of values of accuracy metrics obtained with the approaches used in the selected articles.

Furthermore, Fig. 12 reports the averages of the accuracies detected in the different studies that use the ML algorithm or the DL network (expressed in abscissa). The three best classifiers in terms of average accuracy are, in decreasing order: Cubic SVM and Ensambe (LSTM + RNN) (100%), J48 (99%), and custom classifier (97%).

Figure 12 Average accuracy obtained by the approaches used in the selected articles.

The first classifier, Cubic SVM, is a variant of SVM that uses a cubic kernel function to map data into a high-dimensional feature space. Using a cubic kernel function allows SVM to find more complex and nonlinear decision frontiers in feature space. This can be useful when the data is not linearly separable in the original space and requires a nonlinear representation to be accurately classified, as is often the case in medical/healthcare data.

Ensemble (LSTM + RNN) is a combined neural network i.e. a combination of networks used to improve predictive performance. Both LSTM and RNN are types of neural networks used for sequential processing of data, such as time series or natural language data. Typically, to combine the two networks, one takes an approach of training each model separately on the same dataset and finally combines the predictions using techniques such as averaging or a more sophisticated method such as stacking or fusion.

J48 is a widely used algorithm for creating decision trees. This algorithm requires a labelled dataset, meaning that each example in the dataset is associated with a class label. To select the best attribute, the J48 algorithm splits the dataset at each node of the decision tree allowing for better information gain; the data set is split to optimize the Gini index or gain ratio. The tree construction process continues until a stopping criterion is met. J48 is widely used due to its simplicity, interpretability, and effectiveness in a variety of industries, including the smart healthcare data space. Finally, a custom classifier is a customization of an already existing algorithm, implemented concerning the classification objective. Customization, however, must be done thoughtfully and it is important to evaluate the performance of the customized model through appropriate validations and testing to ensure improvements and avoid unintended errors.

RQ4: What privacy and security issues have taken into account while using health data collected from IoT devices?

The use of health data collected by IoT devices raises significant privacy and security concerns that must be carefully addressed to ensure the protection of people’s sensitive information. For this reason, we also focused our analysis on this topic, evaluating in the selected documents whether privacy or security problems had been found and how these had been addressed during the phases of collection, transmission, and storage of health data by IoT devices.

First of all, we found that the two topics, privacy and security, were addressed in only 29% of the works analyzed. This means that less than one in three articles using health/clinical data questions whether the privacy of users using the devices is respected, considering whether sensitive data is recorded and whether storage techniques comply with data security standards.

In all the documents in which these issues were considered, the problems related to user privacy during the data recording phase by IoT devices were taken into account. Only 13.80% of the works also take into account the issue of data security addressed during the data archiving phase.

These studies also provide treatments aimed at solving these problems. In particular, issues relating to user privacy during the data recording phase(s) have been resolved as follows: considering only some characteristics extracted from the raw data stored and not using the starting data recorded by the device itself (Ware et al., 2022);

when the video recordings of the users filming their faces and voices are also included in the original data, these recordings were eliminated from the analyzed data set (Wu et al., 2022; Cai et al., 2018);

instead of recording user videos, a ‘Radio Frequency Identification’ (RFID) system was used, an automatic digital identification technology that allows the detection of objects, people, and animals, both static and moving, exploiting electromagnetic fields and not video recording (Chen et al., 2022);

the sampling rate is set to 1 KHz to help address privacy issues since speech becomes unintelligible at this sampling rate (Adhikary et al., 2023).

As regards the studies that take into account the issue of data security also during data archiving during and after data analysis, the main solutions adopted for this problem were: the use of edge-fog architectures of the cloud (Chakraborty & Kishor, 2022; Sadhu et al., 2022),

the utilization of encryption techniques to ensure that data collected from IoT devices is encrypted both during transmission and while stored in databases (Sarmah, 2020; Cheikhrouhou et al., 2021),

the adoption of high-security protocols of the storage clouds (Yue et al., 2020),

the conduction of regular security checks and assessments to identify vulnerabilities and risks in the cloud system, plus implement monitoring mechanisms to detect any unauthorized access or suspicious activity (Rahman et al., 2020).

Discussion

Overall, the analysis we conducted highlighted the following results: most of the research articles reviewed deal with ML methodologies rather than DL techniques, and only a small fraction of both;

most articles using ML algorithms use multiple classifiers, while those using DL algorithms typically use a single network, and this pattern remains true even in cases where articles use both approaches;

the most used neural network architectures for IoT smart health problems are (i) CNNs when dealing with neurodegenerative diseases (Prince, Andreotti & De Vos, 2019; Chen et al., 2022), mental problems (Gupta, Bhatia & Kumar, 2021), problems heart problems (Cheikhrouhou et al., 2021), diabetes (Ramazi et al., 2021) and cancer (Masood et al., 2018), (ii) LSTM when it comes to heart problems (Chiang, Wong & Dey, 2021), covid (Neog, Dutta & Medhi, 2022), mental health (Jung et al., 2021), lung disease diseases (Schoun et al., 2018) and walking problems (Staab et al., 2022), (iii) DNNs are mostly used in case of walking problems (Kim et al., 2020), heart diseases (Wu et al., 2022), diseases pulmonary (Wu et al., 2022) and neuro-degenerative diseases (Prince, Andreotti & De Vos, 2019);

the minimum accuracy value obtained is 68%, while the maximum accuracy obtained is 100% and on average the studies reach an accuracy of 88%;

privacy and security issues were evaluated in only 29% of the works analyzed.

Our systematic review reveals a noteworthy emphasis on Machine Learning (ML) methodologies over DL techniques. This inclination can be attributed to several factors. ML algorithms, especially when dealing with traditional classification problems, are often more interpretable. Many researchers treat deep learning methods as an enigmatic system, lacking the means to justify its successful outcomes or make adjustments in instances of classification errors. Interpretability is crucial in healthcare, where clinicians and practitioners need to understand the reasoning behind AI-driven decisions.

In addition, training reliable and effective DL algorithms requires large sets of data. Even though there has been a recent surge in the availability of healthcare data, not all medical applications—especially those concerning rare diseases or events—are ideally suited for deep learning (Ravì et al., 2017).

ML models usually require fewer computational resources compared to DL models, especially those with multiple layers. These DL models often necessitate substantial computing power and have extended processing times.

For what regards DL applications our research work highlights the most implemented architectures. CNNs, for instance, shine when confronted with problems involving image analysis since are adept at detecting intricate patterns in medical images, which are often pivotal in accurate diagnoses. LSTMs, with their sequential data processing capabilities, come to the forefront when tackling issues that involve time-series data. LSTMs can capture dependencies and trends in sequences of health measurements, enabling the prediction of deteriorating health conditions or the detection of subtle changes in a patient’s well-being. This makes them invaluable in real-time monitoring and early intervention. DNNs, on the other hand, are prominently featured in health problems that demand complex feature extraction and data representation. DNNs are characterized by their depth and capacity to learn hierarchical representations, making them suitable for problems that involve multifaceted data sources and complex relationships. Despite exceptional performance offered, the high computational demands associated with DL models can hinder real-time deployment in resource-constrained healthcare environments, where timely interventions are critical. Balancing high accuracy with practical utility and real-world applicability remains an ongoing challenge in smart health research.

Following all the aforementioned considerations, the adoption of DL techniques in a selection of research articles emphasizes their relevance in distinct healthcare contexts. As the volume of data grows, DL technologies can step in where human analysis might falter, enhancing the speed and intelligence of disease diagnosis. This can minimize ambiguity in clinical decision-making. Perhaps the ultimate challenge for deep learning lies in its ability to amalgamate data from various health informatics fields, paving the way for the next era of precision medicine.

A critical avenue for future research revolves around enhancing the privacy and security of health data, as these topics remain insufficiently explored in the literature. With the rapid development and integration of AI technologies into the landscape of smart health, as well as the widespread use of IoT devices for health monitoring, the protection of sensitive patient information has become an essential and paramount concern. One promising research direction involves the development of robust privacy-preserving mechanisms that can safeguard personal health data throughout its lifecycle, from collection and transmission to storage and analysis. Additionally, research efforts should focus on ensuring compliance with evolving data protection regulations and standards in the healthcare sector. By addressing these privacy and security challenges, future research can pave the way for the responsible and effective deployment of AI in smart health, ultimately benefiting both patients and healthcare providers.

Conclusions

Artificial intelligence in healthcare allows us to manage enormous quantities of data, finding connections between information that is apparently unconnected or so numerous that it cannot be analyzed otherwise. The system thus generates new concepts and new associations, for example by detecting, based on the repeated verification of images connected to the diagnostic data, that a certain conformation or colour of the nevi corresponds to given tumour pathologies. It is evident, therefore, that the greater the number of data inserted in the learning phase or during the life of the system, the more accurate and precise the ability to create new connections, see hidden results, and reach conclusions that man would not be able to reach. arrived or would have arrived with extreme difficulty.

In healthcare, artificial intelligence finds space in sectors such as diagnostics, prevention, rehabilitation, telemedicine, robotic surgery, the development of new drugs and vaccines, the organization of services, or the definition of collective behaviour models affecting the assistance.

The availability of wearable devices, capable of detecting important health parameters such as blood pressure, oxygenation, heart rate, or electroencephalogram, is also facilitating telemonitoring and remote assistance. The advantages are evident, allowing forms of control without changing habits or lifestyles and being able to acquire data from the context of the patient’s daily life.

Therefore, the new frontier of wearable devices aims to exploit artificial intelligence not only to provide useful information for the present context, but to anticipate actions, or rather “predict” them to help people even more.

In this regard, we performed an in-depth systematic review of the application of ML and DL technologies to the advanced analysis and prediction of data derived from healthcare IoT devices. The main objective was to contribute to the current state of the literature by exploring the research landscape regarding ML and DL approaches applied in the smart health sector, to highlight challenges and vulnerabilities.

We conducted our research along four main directions and the different analyses conducted first report whether the studies use ML, DL, or both models (i); which models are used specifically (ii); the accuracy and reliability of AI-based models in the smart healthcare landscape (iii); and evaluating important security and privacy issues related to data use (iv).

The results obtained show that there is a more widespread use of ML techniques, rather than DL. We believe this result arises from the fact that ML techniques are generally simpler to implement and interpret than the deep neural networks used in DL. This is particularly important in contexts where the interpretability of models is crucial. DL also requires a significant amount of computational resources and data to effectively train models, which may not always be available. ML techniques, on the other hand, can be less demanding in terms of resources.

We also discovered that in the case of DL the most used networks are CNN, DNN, and LSTM, and that these are chosen for their effectiveness in specific types of problems.

Furthermore, the results demonstrate that the minimum accuracy value obtained is 68% while the maximum accuracy obtained is 100% and on average the studies achieve an accuracy of 88%. This wide range of accuracies reflects the variety of complexity in the problems addressed, the quality of the data available, and the different architectures and methodologies used in the models. A 100% accuracy might indicate cases of overfitting on specific datasets, while such a high average accuracy suggests that, despite the challenges, most studies manage to develop models that perform well. This may be due to the use of advanced validation and optimization techniques, such as cross-validation, hyperparameter tuning, and regularization.

Finally, very few studies consider the issue of data privacy. Many studies are primarily focused on improving model performance, and data privacy may not be an immediate priority. Awareness about the importance of data privacy is growing, but it may not yet be standard practice in all research contexts. Integrating privacy protection techniques, such as data anonymization or differential privacy, can be complex and requires specific skills.

Therefore, this review has helped to clarify several unanswered questions in the context of IOT devices in healthcare, demonstrating the need for further research shortly to establish the use of ML and DL as a viable and mature strategy in the healthcare context.

Supplemental Information

Supplemental Information 1 Raw Data.

Additional Information and Declarations

Competing Interests

Author Contributions

Data Availability

Marina Iammarino is an Academic Editor for PeerJ.

Lerina Aversano conceived and designed the experiments, analyzed the data, performed the computation work, authored or reviewed drafts of the article, and approved the final draft.

Martina Iammarino performed the experiments, analyzed the data, performed the computation work, authored or reviewed drafts of the article, and approved the final draft.

Ilaria Mancino performed the experiments, analyzed the data, performed the computation work, prepared figures and/or tables, and approved the final draft.

Debora Montano performed the experiments, analyzed the data, performed the computation work, prepared figures and/or tables, and approved the final draft.

The following information was supplied regarding data availability:

This is a literature review.

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
