# Peer review of "A systematic review on artificial intelligence approaches for smart health devices"

_PeerJ Computer Science, doi:10.7717/peerj-cs.2232_

## Round 0.1 · original submission · Major Revisions

Dear Authors,

Your article has been reviewed. Based on reviewers' opinions, it needs major revisions before being accepted for publication in PeerJ Computer Science.
More precisely, the following points need to be addressed and resolved in the revised version of the manuscript:

1) The transition from one section to another must be improved to make the article more readable.

2) Overall, the methodology explained is excellent, but the flow of the document in transmitting it to the reader makes it quite difficult to understand. This is also somewhat influenced by the mix of results, which, on the one hand, is positive for the analysis of the review of the results section but makes interpretation difficult when it comes to the methodological section of the review.

3) The authors stated that the manuscript is about a systematic review of artificial intelligence approaches for smart healthcare devices. However, only Machine Learning (ML) and Deep Learning (DL) were mentioned. The authors concluded that only ML and DL are suitable for application to advanced analysis and prediction of healthcare scenarios. How did they arrive at this conclusion in the abstract, starting from a limited number of articles?

4) Figure 1 needs to be redrawn based on a PRISMA flowchart

5) Questions RQ1, RQ2, RQ3, and RQ4 need to be edited and written more clearly. At the moment, they seem to confuse readers. These questions need to be changed, and the application of AI in smart health needs to be explained in detail.

**Language Note:** PeerJ staff have identified that the English language needs to be improved. When you prepare your next revision, please either (i) have a colleague who is proficient in English and familiar with the subject matter review your manuscript, or (ii) contact a professional editing service to review your manuscript. PeerJ can provide language editing services - you can contact us at [email protected] for pricing (be sure to provide your manuscript number and title). – PeerJ Staff

·

Basic reporting

The paper maintains quality to the most parts with some parts required a revisit to work with. Significance is shown to its fullest and rightly done so. The contribution to the field through this paper is required as it unearths some of the unsolved research questions present in the industry that this paper aims to be in. Overall assessment of this paper is great with very few minor changes making it a great read for the readers.
The strengths of the paper is clarity, information depth and use of existing studies to support the claims and objectives done by the authors. It is also relevant to the present world as this tackles some of the issues faced in critical industries like healthcare. The technical knowledge presented in the paper is superior in an easy to understand manner. These strengths make this paper technical as well as knowledgeable when used for the right audience. The weaknesses would be minor hiccups in transitioning from one section to another section particularly in the methodology and results section which can be seen so. There are no other weaknesses in the paper and fixing the only minor change makes this a complete paper for the readers.

Experimental design

Abstract:The abstract starts with the introduction of problem statement through the concepts in terms of domain knowledge. The aim of the paper is explained in a short and crisp manner, reducing the regular length description found in research papers now-a-days. It is followed by the explanation of data collected from the database to the methods and techniques used in processing and evaluation of them. Finally, a vague but short description of the latter sections of the paper is given setting the pace for the reader.
The key points are discussed in short and great keeping the important details for further reading. Clarity is maintained and adequate use of metrics is also done to solidify the paper’s motives and goals. Overall, no changes are required in the abstract part of the paper.


Introduction:
Introduction is broken down into 3 parts where the first part introduces the paper topic with relevant references. Second part consisted of main objectives introduced with goals listed in detail. The final third part consists of a summary of introduction paving the way for other sections.
The authors also gave the paper structure in the last paragraph of the paper with 5 sectional divisions of the paper. This particular inclusion is great as it can make the user more aware about what to expect from the paper step wise.
Overall, the introduction is crisp and to the point by avoiding jargon and inclusion of objectives of the paper and goals in more detail is great. Sectional division of the paper is a welcome addition. Clarity is maintained in the introduction part with

Literature Review:The literature review of this paper is divided into 2 parts with 2nd part having 3 sub divisional parts in it. This shows the granularity presented by the authors in conducting literature review. The first part called as “Related Works” presents some of the classifications done in terms of domain, Artificial Intelligence in general and the pre-existing systems and research on the topic presented by the paper. One of the 2 papers discussed were not following standards set by another paper which this paper also uses as basis and the aspects also been discussed. The remaining studies reviewed by the author revolved around wearable technology, machine learning, and deep learning usage either in collaboration or individually. Finally, reiterating the title of the paper is done but in a more elaborated form with following set standards from the referenced paper.
In the second part’s first division, categorization of AI according to the source in the context of the paper topic is done along with narrowing down the methods into domains. There is also an explanation on applications of the narrowed-down methods. Only downside is that the whole paragraph is a single sentence with commas and semi commas separating in between them. Machine Learning definition is explained which is a good practice to include information on the technique which is used in the paper and also further classification is done extending the background information. Deep learning section is explained with less intensity compared with its previous section. The main topic’s section is short but explained really well with its own examples for better understanding. Establishing the main concept using cues from industry setting some standards is done. In depth information with a good number of references cited showing greater depth in the proposed solution to the topic of the paper. In wearable technology, it is explained well in terms of healthcare use cases.
Overall, the literature review has clarity in it with very few minor hiccups existing in it. The use of studies to support the proposed work is done very well with specification of standards to be used throughout the paper. All the topics were explained in detail and crisp format showcasing the quality work done by the authors towards conducting a literature review. All the relevant sources are cited properly with critical analysis done sufficiently.

Methodology:The methodology is termed as research and also continues into the result section of the paper which might sometimes confuse the reader but nonetheless presents good insights on the topic at hand. Research approach is given based on the referred paper which is taken as base for this paper. These research questions are defined in detail giving clarity on the objectives of the paper. To support this further, not only references but the dataset’s database is also introduced and explained in detail with further exploratory data analysis (EDA). There are no major summary points or paragraphs for the research section as it flows into the results sections which is to be noted.
For the review, this flow is considered to be under this section of the review where the methods discussed in the literature review were used and listed under domains which simplifies the analysis process. This is a great work done by the authors in condensing the outputs to the readers to understand.
Overall, the methodology explained is great but the flow that the paper had in conveying this to the reader makes it somewhat difficult to understand. This also gets a bit affected with the mix of results which is good on one hand for the results section review analysis but makes it difficult to interpret when it comes to the methodology section of the review.

Validity of the findings

Results:When it comes to results, it is explained very well with the help of tables and analytical figures showing the effective research work and implementation done by the authors. It is also further broken down into research questions used as sub-sections to depict the results obtained. The quality of the results obtained is great as it is both supported by the literature review as well as the standards defined from the basis paper. Each figure is explained in detail revealing insights which are translated from figures to words which are simpler to understand. After the explanation of a research question through results is done, a comprehensive summary of the section is done using the results obtained, references from the paper and comparative points are formed to summarize the section. Overall, the results section looks nice and requires no further changes in it.

Discussion:The discussions start with the highlights from the results section paving way for deeper discussion points. Lot of referencing is done to explain all the important aspects of the result findings. The opinions held by the authors are also explained in detail with relevant information from literature review, methodology, and results sections.
The discussion section also contains the future works as the last paragraph with properly laid out information of the paper’s future. This reduces more sections of paper and flow being maintained through future works being presented as the evolution of future works from all the previous sections and paper as a whole.
There is less talk on the limitations of the paper even though it is acknowledged vaguely. Nonetheless discussion sections presented a more comprehensive approach in showcasing the findings and connection of it with other sections of the paper.

Conclusion:Conclusion of the paper is really good with key findings being specified greatly with effect and also the implications of the results and future work discussed is good. Mixture of all the sections summary is done and interlinked greatly with concepts supporting the findings. They concluded that the research questions defined are answered and demonstration of the methods gave insights on topics strategy development for future works. To the end, conclusion is in detail and gives an all round summary of the paper from inception to finish of the paper.

Additional comments

Clarity and Organization: The paper for the most part is organized well and has clarity in all sections. Some of the sections which could be made better are methodology and results in terms of having the transition of content from one section to another. Writing style is appropriate for the intended audience as it is easy to understand for the most part and also small things like abbreviations and technical terms were handled great. The idea of the paper is presented very well and is used whenever it was necessary to specify it for further explanation. Overall the clarity is maintained and by minor changes in the organization makes this paper really great for the intended readers.

Improvements were specified in their respective sections of the review. Please refer to them for any revisions or improvements.


References:References taken were great as they are from reputable sources showing the quality of the references used. In terms of citations, the style of citing references is done great. It is also to be noted that all the references are made in alphabetical order adhering to the paper style guide. One suggestion can be made which is to use the hyperlinks to link the citation and reference paper to help further finding of the reference for the reader.

·

Basic reporting

Even though the strategy chosen for the review is appreciable, it fails to fill some gaps.
Since it discusses recent developments and the article was submitted in 2024, it should be concentrated until at least 2023. However, the articles were taken up to 2022. Is there any specific reason for this?

Experimental design

Are any specific standards followed up, like PRISMA, for structuring the survey?
Is there any specific purpose for restricting the literature to IEEE and Elsevier only? Some primary indexing sources, such as Web of Science, PubMed, Scopus, or a combination, would have been considered.

Validity of the findings

For your information, the "Editor " mentioned should be specified as "Editorial."
Figure 4 shows the distribution of IEEE editorial, but no such distribution is found in Elsevier taken up for study.
It should not be restricted to only two editorials (or specify the same in the title), including all sources; exclusion criteria might be included with a valid reason for considering only these two editorials.
Data derived and summarised in Table 2 should be submitted as supplementary data with all raw data abstracted from the original research articles.

·

Basic reporting

This article depicts a systematic review on Artificial intelligence approaches for Smart Health Devices
The abstract has to be changed, the subject and scope of the research has to be included.
Searching criteria, details about the scientific databases, article selected has to be included in the Relevant Articles section. Rather the abstract includes all these sections. These contents has to be removed from the abstract.
The authors mentioned that that the manuscript is about a systematic review on Artificial intelligence approaches for Smart Health Devices. But only the Machine Learning (ML) and Deep Learning (DL) were mentioned.
The authors came to a conclusions that only ML and DL are suitable for application to advanced analysis and prediction of healthcare scenarios. How do they came to this conclusion in the abstract itself by from a limited number of articles ?
Related works section has to be enhanced by including more recent literatures on the applications of Artificial intelligence based methodologies'.
All the steps in systematic reviews and meta-analyses has to be followed.

Experimental design

Figure 1 has to be re drawn based on a PRISMA flowchart
More sections should be included regarding the Artificial intelligence approaches for Smart Health Devices from recent literature
The results section is described in many sections. Rather, the main scope of the research: Application's of AI in Smart health has to be explained in detail
Meta-analysis part has been explained throughout the manuscript. It has to be limited and the the main scope of the research: Application's of AI in Smart health has to be explained in detail

Validity of the findings

Metanalysis part has been explained throughout the manuscript. It has to be limited and the the main scope of the research: Application's of AI in Smart health has to be explained in detail
Conclusion shas to be crisp and clear
The questions RQ1, RQ2, RQ3 and RQ4 has to be changed and written more clearly. At present, these seems to be confusing the readers.
These questions has to be changed and the Application's of AI in Smart health has to be explained in detail.
The authors have selected a limited papers and came to a conclusion about the distributions of the number of algorithms used in those selected articles.
Many recent and related articles has to be selected during this process. This process has to be included through the PRISMA flowchart.

Additional comments

This manuscript has to be modified based on the basic systematic reviews and meta-analysis procedure.

---

## Round 0.2 · accepted · Accept

Dear Authors,

Based on the re-review and my own reading, your manuscript has been accepted for publication in PeerJ Computer Science.

·

Basic reporting

No comments

Experimental design

No Comments.

Validity of the findings

No comments.

Additional comments

The Research paper looks good after the revision made good progress and accepted to be published.

Best revisions:-
1- The Abstract improved and explained clearly
2- The methodology is explained well.
3- All the Questions were explained clearly.
4- The results are explained very well with the help of tables and analytical figures showing the practical research work and implementation done by the authors.

While publishing the article, please see the comments below.
1- Ensure all sources are correctly cited using a consistent and recognized referencing style.
2- Proofread the article thoroughly to address typos, grammatical errors, or inconsistencies.

Thank you all.